# The Prisoner's Dilemma paradigm provides a neurobiological framework for the social decision cascade

Khalil Thompson[1]*, Eddy Nahmias[1], Negar Fani[2], Trevor Kvaran[1], Jessica Turner[1], Erin Tone[1]

**1** Department of Psychology, Georgia State University, Atlanta, Georgia, United States of America,
**2** Department of Psychiatry and Behavioral Sciences, Emory University, Atlanta, Georgia, United States of America

* kthompson70@gsu.edu

**Data Availability Statement:** All relevant data are within the paper and its Supporting Information files.

**Funding:** This work was supported by: a National Science Foundation Graduate Research Fellowship

## Abstract

To function during social interactions, we must be able to consider and coordinate our actions with other people's perspectives. This process unfolds from decision-making, to anticipation of that decision's consequences, to feedback about those consequences, in what can be described as a "cascade" of three phases. The iterated Prisoner's Dilemma (iPD) task, an economic-exchange game used to illustrate how people achieve stable cooperation over repeated interactions, provides a framework for examining this "social decision cascade". In the present study, we examined neural activity associated with the three phases of the cascade, which can be isolated during iPD game rounds. While undergoing functional magnetic resonance imaging (fMRI), 31 adult participants made a) decisions about whether to cooperate with a co-player for a monetary reward, b) anticipated the co-player's decision, and then c) learned the co-player's decision. Across all three phases, participants recruited the temporoparietal junction (TPJ) and the dorsomedial prefrontal cortex (dmPFC), regions implicated in numerous facets of social reasoning such as perspective-taking and the judgement of intentions. Additionally, a common distributed neural network underlies both decision-making and feedback appraisal; however, differences were identified in the magnitude of recruitment between both phases. Furthermore, there was limited evidence that anticipation following the decision to defect evoked a neural signature that is distinct from the signature of anticipation following the decision to cooperate. This study is the first to delineate the neural substrates of the entire social decision cascade in the context of the iPD game.

## Introduction

Many of the consequential decisions people make in their day-to-day lives occur within social contexts. Social decision-making involves interdependency and mutual commitment; individuals must consider not only how possible outcomes will affect them, but also how those

[DGE-1550139] to K. Thompson, a GSU Brains &
Behavior seed grant to E. Tone & E. Nahmias, the
GSU/GA Tech Center for Advanced Brain Imaging
grant to E. Tone & J. Turner, a National Institute of
Health grant to J. Turner [NIH-5R01MH121246],
as well as a 2CI Fellowship for Neuroimaging from
GSU awarded to K. Thompson. The funders had no
role in study design, data collection and analysis,
decision to publish, or preparation of the
manuscript. https://nsf.gov/ https://www.nih.gov/
https://neuroscience.gsu.edu/brains-behavior/
http://www.cabiatl.com/CABI/.

**Competing interests:** The authors have declared
that no competing interests exist.

outcomes will affect other people with similar or conflicting needs and desires [1]. Functional magnetic resonance imaging (fMRI) studies that have modelled social interaction using the Prisoner's Dilemma (PD) paradigm have typically targeted neural activity during isolated snapshots of the process of either making a decision or receiving feedback about one's decision [2–6]. The unfolding of the interaction from decision-making, to the anticipation of outcome, to receiving feedback on one's decision constitutes a "cascade" of events that may be better understood as a dynamic, cyclical flow of interdependent social phenomena. The purpose of the present paper is to provide an integrated picture of the neural mechanisms of the entire "social decision cascade". Of note, we include the anticipatory process that occurs between decision-making and processing of outcome feedback, which has received minimal attention in the literature.

The PD task is an economic-exchange game that elicits distinct, quantifiable patterns of interaction (e.g., displays of pro-social, submissive, hostile or competitive behavior) in a structured context that models reciprocal altruism and strategic conflict [6, 7]. In this task, an individual and a social partner engage in a series of bilateral exchanges that can yield rewards or punishments, depending on the choices that each individual makes. Each exchange in the task constitutes a round that unfolds as a series of <u>three</u> phases. First, the participant makes a dichotomous **decision** (to cooperate with or defect from the other player), then the participant waits in **anticipation** of the outcome (whether the co-player chooses to cooperate or defect), and finally, the co-player's response is revealed and the participant receives **feedback** regarding the monetary reward that results from the conjunction of the two players' decisions.

The principles underlying the application of economic-exchange tasks to the study of social behavior stem from game theory, which describes how people navigate strategic interactions and bargaining scenarios while aiming to optimize or maximize their interests by selecting options that provide the greatest personal utility [8, 9]. However, the current literature indicates that game theory and rational decision-making models cannot completely account for all of human behavior when social norms, preferences, and situational context are taken into consideration [10, 11]. We hope that by delineating neural mechanisms of decision-making in this reciprocal exchange-based task we can help to further refine theoretical models that illustrate how humans adapt their behavior within diverse social contexts.

To date, at least 42 fMRI studies have used the PD task to characterize the neural correlates of social behavior during individual social decision-making phases (e.g., decision or feedback) in samples drawn from diverse populations. Of these 39 studies, 34 used the iterated format (iPD), which allows for repeated interaction with a co-player; the remaining eight used the one-shot format, in which a participant engages in only one round with different co-players. Of the 33 iPD studies that we located, only seven targeted healthy populations and developed hypotheses solely focused on the neural substrates of PD behavior and gameplay in healthy populations. The remainder focused on clinical populations or groups subjected to external environmental constraints such as the endorsement of social preference, priming about the reputation of co-players, and the administration of exogenous chemical substances (e.g., oxytocin or vasopressin). Therefore, we will direct our attention to these seven studies for neurobiological evidence of distinctive PD gameplay.

Four of these studies investigated the neural basis of social cooperation with human co-players [2, 3, 11, 12]. Findings from the earliest of these studies indicated that decision-making in the iPD game was associated with activity in the rostral anterior cingulate (rACC) and the caudate [2]. This study and two subsequent studies also found activity in two additional networks when participants received feedback about their co-player's decision [2, 3, 12]. Co-player cooperation was positively correlated with activity in the orbitofrontal cortex, the rACC, and the ventral striatum [2, 12], both of which have been shown to respond during

subjective valuation [13, 14]. In contrast, co-player defection was positively correlated with activity in the amygdala, anterior insula, ACC, and hippocampus [3, 12], a set of structures implicated in fear-based associative learning [15, 16].

The last study examined neural activity averaged across the entire iPD task rather than during isolated phases of decision-making [11], and focused on whether BOLD signals varied as a function of the co-player's ostensible playing style. Gameplay with "cooperative" partners preferentially evoked activity in the valuation network. In contrast, gameplay against "competitive" partners (those with a greater tendency to defect) evoked activity in the temporoparietal junction (TPJ) and the precuneus, brain regions implicated in understanding others mental states and self-referential processing respectively [17, 18].

Given the increasing breadth of the iPD/neuroimaging literature, it is surprising that, to date, no study has analyzed brain function during anticipation of the outcome following decision-making in the iPD. Such analyses seem important, given evidence that prior expectations about the consequences of a decision can influence how an interaction progresses [19, 20]. Moreover, research examining gain or loss in monetary and social reward situations has yielded evidence of a possible anticipatory processing network that comprises the dorsomedial prefrontal cortex (dmPFC), the anterior mid cingulate [21], the anterior insula [22] and the striatum [23].

We defined anticipation in the context of this study, as the period of time between a decision and an incentivized outcome. This period of time, during which a person presumably generates expectations about the outcome, has been linked in cued response studies to "anticipatory affect", which can encompass a broad spectrum of emotions [24]. Cued response studies allow for the examination of anticipation as an independent construct that is dissociated from, but still coupled with the decision-making process [25–27]. Cued response paradigms contrast with paradigms such as the mixed gamble task, in which anticipation and decision-making are modelled as a nondissociable combined unit [28–32].

The purpose of the current study was to characterize neural correlates of cognition and behavior during the decision-making, anticipation, and feedback phases of the iPD game in a healthy sample. We localized markers of neural activity in each phase and compared activity between phases. We predicted, based on previous findings that a network including the dmPFC, caudate, aMCC, and TPJ would activate preferentially during the decision-making phase of the task. Furthermore, a network spanning the vmPFC/OFC, rostral ACC/aMCC, TPJ, ventral striatum, anterior insula, amygdala and hippocampus would be significantly activated during processing of feedback. Although predictions about anticipation were necessarily more speculative due to the sparse literature on this topic, we predicted that the dmPFC, rostral ACC, the anterior insula and the striatum would show significant activation during the anticipation phase of the task.

## Methods

### Participants

The data for this study were drawn from two independent datasets. One set was collected at the Georgia State University/Georgia Tech Center for Advanced Brain Imaging (CABI) in 2016 and one was collected at the Emory University Biomedical Imaging Technology Center (BITC) in 2008. Both datasets were collected using identical behavioral paradigms and subject recruitment procedures.

The Georgia State University and Emory University Institutional Review Board (IRB) reviewed and approved the above referenced study in accordance with 45 CFR 46.111. The IRB has reviewed and approved the study and any informed consent forms, recruitment

materials, and other research materials that are marked as approved in the application. Written informed consent was obtained for this study.

For the 2008 dataset, 19 subjects were scanned; however, usable data from only 14 subjects were available. Data for the remaining five subjects had been corrupted during storage and could not be recovered. Subjects were recruited from the GSU undergraduate psychology student pool via the SONA online questionnaire system. Participants were scanned at the Emory University BITC. For the 2016 dataset, 20 subjects were recruited using the methods from the 2008 study. All 20 were scanned at the CABI; however, data from only 17 subjects were usable (two subjects exited the study prematurely due to elevated anxiety and one subject failed to remain engaged with the task for substantial periods of scan time). In total, we had complete and usable data from 26 females and 5 males, with a mean age of 20.6 years (SD = 3.5 years; see Table 1 for detailed demographic comparisons of the datasets).

## Experimental design

The experimental procedure was identical for the two datasets. Following consent, an examiner informed participants that they would play a 20-round game three times with different study participants via a wireless computer network. Confederates completed consent and training procedures with the actual participant, whom the examiner then selected from the group (apparently at random) to play the game in the scanner, while the others ostensibly played the game in separate rooms. During each of the three games that constituted a session, two players (the participant and a computerized co-player that the participant was deceived into believing was a real human) independently chose, during each of the 20 rounds, to cooperate with or not cooperate with each other. After both players submitted their choices, the outcome of the round appeared on the screen, along with a running total of each player's cumulative earnings for a game. Periodically during and after the game, participants were asked (via the computer screen) about their perceptions of and predictions about their co-player's intentions and goals, as well as about their own emotional responses during play and their levels of confidence in their predictions; these self-report data are not included in the present manuscript. A task overview, as well as timing information during scanning, is presented in S1 Fig.

At the end of the study session, in accordance with guidelines for ethically appropriate authorized deception [33], the examiner debriefed participants about the deception involved in the task and the motivation for its use. No participants expressed concerns. All subjects reported being deceived and thus their data were included in subsequent analyses (see S1 File).

**Table 1. Demographic data for participants from the CABI and Emory sites.**

| | CABI | | | Emory | | |
|---|---|---|---|---|---|---|
| | All (N = 17) | Female (N = 15) | Male (N = 2) | All (N = 14) | Female (N = 11) | Male (N = 3) |
| | M (SD) | M (SD) | M (SD) | M (SD) | M (SD) | M (SD) |
| **Age** | 20.4 (2.5) | 19.8 (1.5) | 25.0 (4.2) | 20.6 (4.6) | 20.6 (4.9) | 20.7 (3.8) |
| | N (%) | N (%) | N (%) | N (%) | N (%) | N (%) |
| **Ethnicity** | | | | | | |
| White | 7 (41) | 3 (20) | 1 (50) | 7 (29) | 6 (55) | 1 (33.3) |
| African-American | 4 (29) | 6 (40) | 0 (0) | 5 (43) | 4 (36) | 1 (33.3) |
| Hispanic | 1 (.06) | 2 (13) | 0 (0) | 1 (14) | 1 (9) | 0 (0) |
| Asian-American | 4 (24) | 4 (27) | 1 (50) | 1 (14) | 0 (0) | 1 (33.3) |

## Task design

In each 20-round iPD game [2] rounds proceeded as shown in Fig 1; the participant chose to cooperate or not cooperate, and then waited for a "co-player", who independently decided to cooperate or to not cooperate (defect). The participant and co-player were equally rewarded (Reward payoff—**R**; $2) if both cooperated; if one player defected but the other cooperated, the betraying player received a reward (Temptation payoff–**T**; $3) while the cooperating player received nothing (Sucker's Payoff–**S**; $0). If both chose to defect, both received a diminished reward (Punishment Payoff–**P**; $1) [34].

The monetary distributions depicted in Fig 1 are organized to conform to the universal scaling parameters for the PDG as an evolutionary dyadic game that promotes cooperation through a number of different reciprocity mechanisms [34–37]. In order to maintain universal dilemma strength in both limited and unlimited well-mixed populations and construct the necessary parametric constraints for the PDG, the gamble-intending dilemma ($D_g$') and the risk-averting dilemma ($D_r$') must be equal and greater than 0 such that: ($D_g$' = (T-R)/(R-P)) and $D_r$' = (P-S)/(R-P)) [34–37]. This generates a Donor & Recipient dilemma template where, given a single decision, defection is incentivized at no cost to the defector; however, given repeated interactions, cooperation is incentivized but at a cost to the cooperator, who risks betrayal and an omission of a reward for the current round [38].

All participants played three games—in two, they were deceived to believe that they were playing with a human confederate and shown a picture of that confederate before starting the game (but they actually played a computer algorithm) and in one they were told that they were playing a computer program. The order of the three games was randomized for each participant (see S2 File).

The participant was given up to six seconds to make a decision in each round. The decision was followed by a 3-, 6-, or 9-second jittered interstimulus interval which constituted the anticipation phase of the round. After the jitter period, feedback regarding the round outcome was presented for six seconds. See S3 File.

The 20-round game was split into four 5-round blocks, with an additional blank round included in each block. After every five rounds, the participant was given an indefinite amount of time to answer four questions (two about their feelings, two about their assessment of the co-player's intentions) before beginning the next 5-round block. After the last 5-round block of the game, the participant answered four final emotional assessment questions and then viewed their total earnings for the game. After 12–20 seconds, the participants then answered 10 additional questions about their experience of play. Each game proceeded in this fashion. Participants were paid the average of the amount that they earned over the three games.

Out of the 40 rounds that each participant played with the "human" confederate, an average of 12 rounds resulted in mutual cooperation (CC); 6 resulted in unreciprocated cooperation (CD); 9 resulted in unreciprocated defection (DC); and 13 resulted in mutual defection (DD). Out of the 20 rounds played with the computer, an average of 4 rounds resulted in CC; 3 resulted in CD; 5 resulted in DC; and 8 resulted in DD. A contingency table detailing total frequencies of cooperate and defection for the human and computer games is displayed in Table 2.

## Scanning procedure

**2008 data.** The 2008 dataset was collected using a Siemens TIM Trio 3T MRI scanner equipped with a 12-channel head coil. E-Prime 1.1 was used to present task stimuli (Psychology Software Tools, Inc.). Participants recorded decisions to cooperate or defect using a handheld, 4-button response box. A localizer and a manual shim procedure preceded each

## Decision

## Anticipation

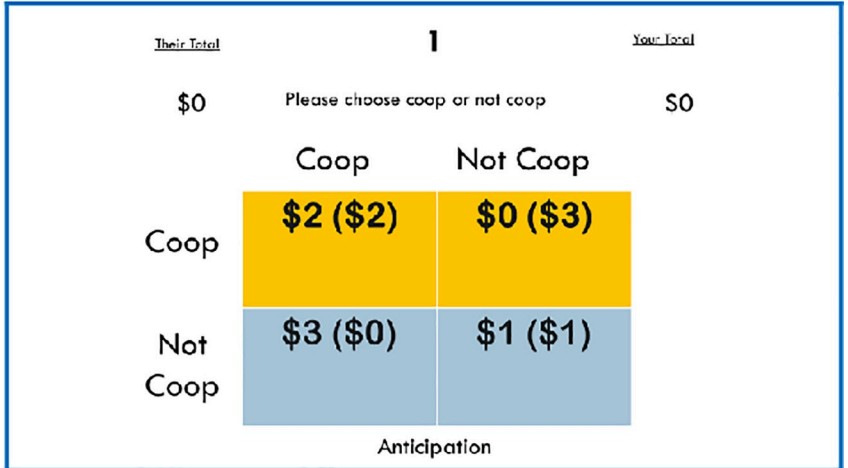

## Feedback

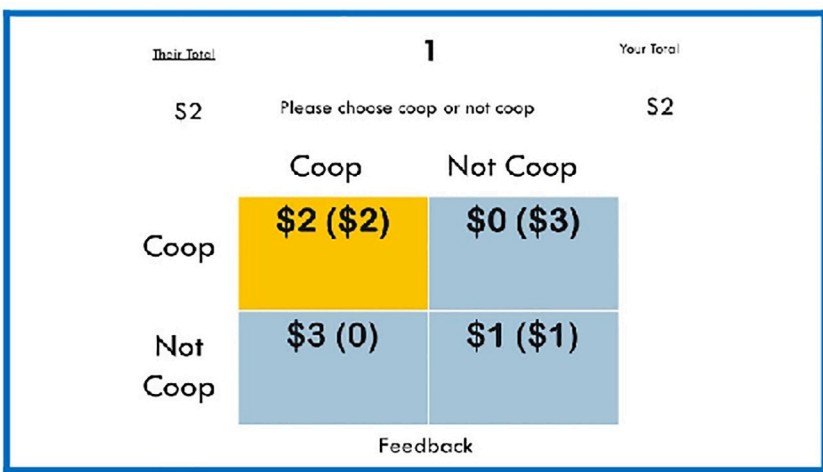

**Fig 1. An example of a mutual cooperation round (CC) during the iterated Prisoner's Dilemma game.** Each round comprises decision, anticipation, and feedback phases of the task. The participant's choices are located on the left of the 2x2 payoff matrix while the co-players choices are located a the top of the matrix.

functional scan. A functional task-related blood-oxygen-level-dependent (BOLD) scan was acquired with a ZSAGA functional protocol. ZSAGA is a method for reducing the influence of susceptibility artifacts in echo planar imaging [34]. The number of volumes varied depending on time spent on task (participants spent variable amount of time completing emotional assessment questions); TR = 3,000 ms; TE 1 = 30 ms; TE 2 = 65.8ms; matrix size = 64 x 64mm; FA = 90˚; 3.3 x 3.3x 3.3 mm$^3$ voxels; 30 interleaved slices; FOV = 210 mm. A high-resolution anatomical image was also acquired using a T1-weighted standardized magnetization gradient echo sequence to aid spatial normalization (MPRAGE; sagittal plane; TR = 2300 ms; TE = 3.02 ms; matrix size = 256x256 mm, 1 mm$^3$ isomorphic voxels, 176 interleaved slices; FOV = 256 mm; flip angle 8˚).

**2016 data.** The 2016 dataset was collected using a Siemens TIM Trio 3T MRI scanner equipped with a 12-channel head coil. E-Prime 2.0 was used to present the task stimuli (Psychology Software Tools, Inc.), and the responses were collected using a Current Designs MRI compatible button box. A localizer and a manual shim procedure preceded each functional scan. A functional task-related BOLD scan was acquired with a T2*-weighted echo-planar functional protocol (number of volumes vary depending time spent on task; TR = 2,000 ms; TE = 30 ms; matrix size = 64 x 64mm; FA = 77˚; 3.4 x 3.4x 4.0 mm$^3$ voxels; 33 interleaved slices; FOV = 220 mm). A high-resolution anatomical image was also acquired using a T1-weighted standardized magnetization spoiled gradient echo sequence to aid spatial normalization (MPRAGE; sagittal plane; TR = 2250 ms; TE = 4.18 ms; GRAPPA parallel imaging factor of 2; matrix resolution size = 256x256 mm, 1 mm$^3$ isomorphic voxels, 176 interleaved slices; FOV = 256 mm; flip angle 9˚).

## Preprocessing

**2008 data.** Using Statistical Parametric Mapping (SPM)12 (Wellcome Trust Center for Neuroimaging), the 2008 functional data were corrected for slice timing and motion, realigned and registered to the mean image, spatially normalized to the SPM Montreal Neurological Institute (MNI) template and resliced into isotropic 2mm voxels, and smoothed using an 8mm FWHM Gaussian kernel.

**2016 data.** Using Data Processing Assistant for Resting-State fMRI (DPARSF) software [35], each subject's functional data were corrected for slice timing and head motion, and co-registered to their anatomical data. The images were resliced and resized to match the scale and dimensions of the original 2008 dataset, then spatially normalized to the SPM MNI

**Table 2. Cooperation, defection and outcome rates in the human and computer games.**

| Human (40 rounds) | | | Participant | |
|---|---|---|---|---|
| **Co player** | | | Cooperate (M/Range) | Defect (M/Range) |
| | | Cooperate | 12 (3–31) | 9 (4–17) |
| | | Defect | 6 (1–13) | 13 (3–26) |
| **Computer (20 rounds) Participant** | | | | |
| **Co Player** | | | Cooperate (M/Range) | Defect (M/Range) |
| | | Cooperate | 4 (0–14) | 5 (2–12) |
| | | Defect | 3 (0–8) | 8 (3–19) |

template and smoothed using an 8mm FWHM Gaussian kernel. The quality of the co-registration and normalization procedure was evaluated by visually inspecting the fMRI images for any inconsistencies.

## Behavioral analysis

To compare participants' average cooperation rates during gameplay across co-players we conducted a one-way repeated measures ANOVA with co-player (human, computer) as the within-subjects factor and cooperation rate during the human and computer games as the dependent variable.

## Neuroimaging analysis

General linear modeling (GLM) in SPM12 was used to estimate event-related BOLD response amplitudes relative to baseline (periods of the minimal task engagement between the phases) across the three phases of the task at the individual subject level and the group level. Primary regressors included two regressors for the decision phase, two regressors for the anticipation phase, and four regressors for the feedback phase of the task, as listed in Table 3.

To account for unrelated cognitive processes that could confound results, a regressor was included for the time points at which participants answered emotional assessment questions. Further, to account for the fact that two out of the three games were played against a "human" and one game was played against a computer, the regressors included a set that distinguished between rounds played against a human and a computer. In total, we included 18 task regressors (9 human and 9 computer regressors) in our design matrix (see S2 Fig). Finally, we included a framewise displacement (FD) regressor in the single subject analyses as an additional motion nuisance covariate.

Two-tailed one-sample t-tests contrasted activity within each individual human regressor versus baseline. Based on our expectation that participants would respond differently to cooperative vs. uncooperative (monetary gain/loss) and reciprocated vs. unreciprocated (social coordination/conflict) feedback, we collapsed round types as follows: CC and DC (Co-Player Cooperation), CD and DD (Co-Player Defection), CC and DD (Reciprocated), and CD and DC (Unreciprocated). We took this approach to increase the power of the analysis and to permit distinct evaluation of responses to the social conflict and monetary gain and loss, consistent with previous PD research [3, 36, 37]. Additionally, direct contrasts were used to compare BOLD responses within phases [(Ex. (CC + DD) > (CD + DC), etc.] and between phases (Ex. Decision > Feedback etc.).

A total of sixteen group-level analyses were conducted. Site was included as a covariate in all group-level analyses (see S3 Fig in the supplemental materials Section 6 for a visual overlay

**Table 3. Description of task regressors used for fMRI analysis.**

| Task Regressor | Symbolic Representation |
| --- | --- |
| Decision to Cooperate | Decision (C) |
| Decision to Defect | Decision (D) |
| Anticipation following Cooperation | Anticipation (C) |
| Anticipation following Defection | Anticipation (D) |
| Mutual Cooperation | CC |
| Unreciprocated Cooperation | CD |
| Unreciprocated Defection | DC |
| Mutual Defection | DD |

of brain activity between the two sites). All results were corrected for multiple comparisons using familywise error rate correction (FWE), and the significance threshold was established at $p < .05$, with a spatial extent threshold of 30 mm$^3$. Results for the decision and feedback contrasts surpassed a t-statistic of 6.05 ($df = 30$).

Results for the anticipation phase contrasts did not survive FWE correction, with the exception of activity in the occipital lobe. We conducted exploratory analyses using an uncorrected voxel-wise primary threshold set at $p < .001$ and a cluster-wise FWE-corrected threshold determined by SPM12 [38]. Because direct contrasts also did not survive FWE correction, we used the same cluster-based threshold method to conduct exploratory analyses on these contrasts (e.g., Decision Coop>Decision Def, CC+DD<CD+DC, etc.) within and between phases of interest.

## Results

### Behavioral analysis

There was a significant effect of co-player on the average cooperation rate between games, $F(1,30) = 14.37$, $p < 0.001$. Overall, participants tended to cooperate more against "human" co-players (M = 45.88, SD = 17.78) than they did against computer co-players (M = 34.12, SD = 19.47; see Fig 2).

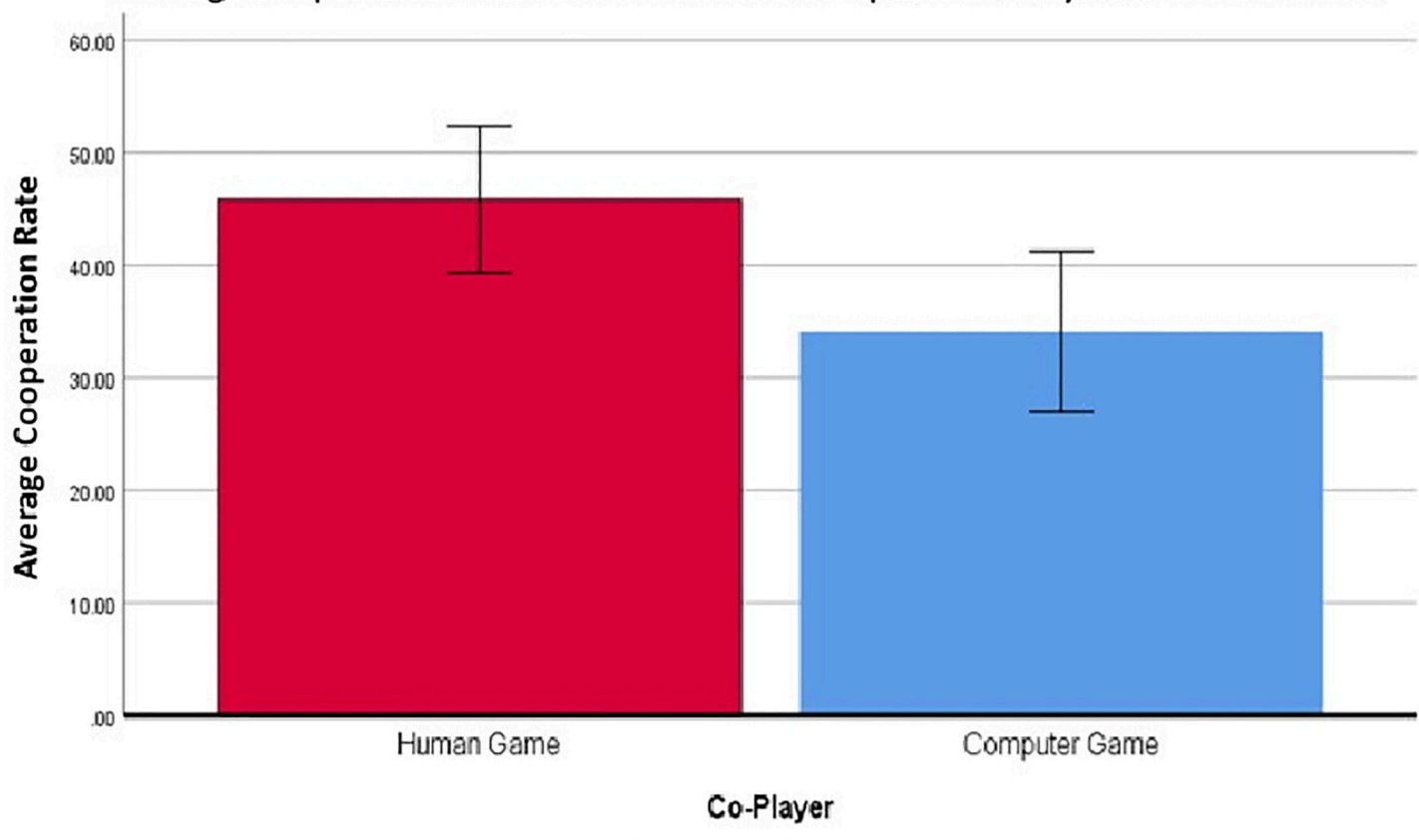

**Fig 2. A graphical illustration of the participants' average cooperation rate against "human" and computer co-players in the PD.** Participants tended to cooperate more often against their human co-players suggesting that prosocial norms were at least partially taken into consideration during these games.

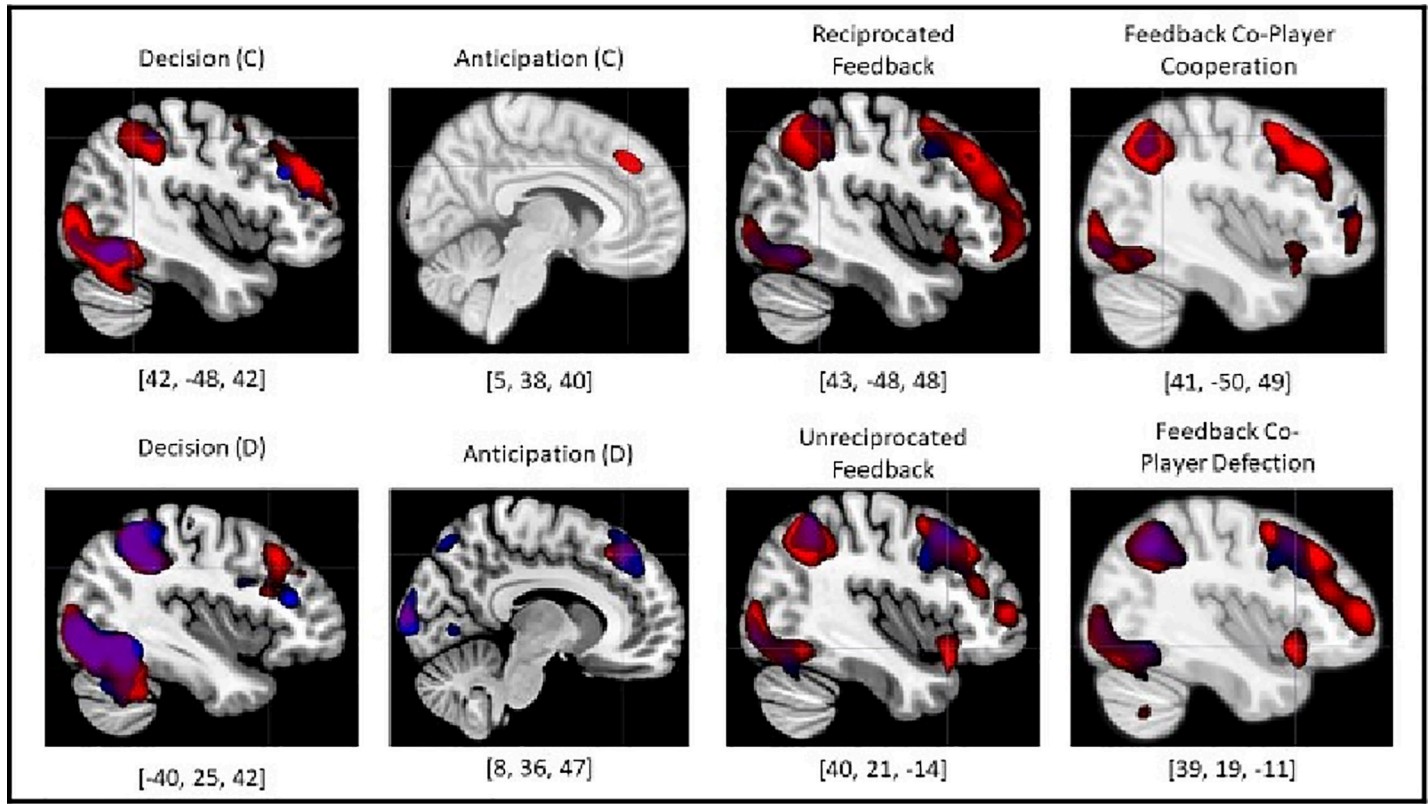

**Fig 3. BOLD activity during all phases of the iPD task.** In each contrast map the cursor is placed within the peak voxel of activity for each condition. The dmPFC and the bilateral TPJ were the only regions that were recruited across all three phases of the task, suggesting that the PD task serves as a valid and robust model of social reciprocation despite its underlying economic principles. (Decision and feedback phases are thresholded at t(29) = 6.00, p < .05; FWE-corrected voxel-wise threshold while the anticipation phase was thresholded at t(29) = 3.40, p < .001 uncorrected voxel-wise threshold; FWE-corrected cluster-wise threshold determined by SPM12). (**Red = Human Game, Blue = Computer Game, C = Cooperate, D = Defect**). Ex. [42, −48, 42] = peak voxel MNI coordinates.

### Human game neuroimaging baseline contrast analysis

**Decision phase.** During the decision to cooperate, significant activation relative to baseline was detected in diverse prefrontal cortical regions of the brain (see Fig 3)—the lateral OFC, bilateral ventrolateral prefrontal cortex (vlPFC), and dmPFC/aMCC. Significant activity was also identified across the parietal lobe, including the left TPJ, bilateral superior parietal lobule and precuneus. Lastly, significant activation was detected in the bilateral anterior insula and the bilateral hippocampus (see Table 4).

During the decision to defect, significant activation was again elicited in regions across the prefrontal cortex (see Fig 3). Lateral OFC and bilateral vlPFC activity were consistently active during decisions to cooperate and to defect. However, in contrast to activation patterns observed during cooperation decisions, decisions to defect were positively correlated with right dlPFC activity, but not dmPFC/aMCC activity. Significant activation to that observed during cooperation decisions emerged in the parietal lobule, including left-lateralized TPJ activity. Bilateral hippocampus activity was present; however, in contrast to what was observed during the decision to cooperate, a significant neural response was not elicited in the bilateral anterior insula (see Table 4).

**Anticipation.** During the anticipation phase, no individual voxels survived FWE correction in a whole brain voxel-wise analysis, therefore we report results were obtained using a cluster-wise thresholding approach and FDR-correcting significant clusters. During

**Table 4. Significant regional activation during decision phases.**

| Name of Region | Brodmann Area | Voxels | MNI Coordinates | | | t(29) | p-value (p < .05; FWE-corrected) |
|---|---|---|---|---|---|---|---|
| | | | x | y | z | | |
| Decision (C) | | | | | | | |
| R dorsomedial PFC | 8 | 46 | 6 | 26 | 46 | 7.12 | .001 |
| L lateral OFC | 11 | 14 | -42 | 44 | -14 | 7.37 | .001 |
| R ventrolateral PFC | 45 | 410 | 45 | 44 | 25 | 10.82 | .001 |
| L ventrolateral PFC | 45 | 91 | -48 | 38 | 22 | 8.02 | .001 |
| R ant midcingulate | 32 | 135 | 6 | 29 | 34 | 7.92 | .001 |
| L temporoparietal junction | 40 | 441 | -36 | -46 | 40 | 11.26 | .001 |
| R inf parietal lobule | 40 | 222 | 48 | -37 | 46 | 8.46 | .001 |
| R sup parietal lobule | 7 | 186 | 27 | -64 | 40 | 9.86 | .001 |
| L sup parietal lobule | 7 | 212 | -24 | -67 | 40 | 10.04 | .001 |
| Precuneus | 7 | 243 | 12 | -70 | 52 | 9.55 | .001 |
| L ant insula | 48 | 24 | -39 | 14 | -5 | 7.16 | .001 |
| R ant insula | 47 | 14 | 36 | 17 | -8 | 6.50 | .001 |
| R thalamus | | 27 | 18 | -10 | 1 | 7.88 | .001 |
| L hippocampus | 27 | 151 | -21 | -31 | -2 | 9.29 | .001 |
| R hippocampus | 37 | 86 | 24 | -28 | -2 | 9.34 | .001 |
| Occipital lobe/cuneus | 17 | 2864 | 15 | -97 | 10 | 13.43 | .001 |
| Decision (D) | | | | | | | |
| R dorsolateral PFC | 9 | 69 | 18 | 56 | 31 | 9.77 | .001 |
| R lateral OFC | 11 | 27 | 30 | 56 | -11 | 7.07 | .001 |
| R ventrolateral PFC | 45 | 146 | 42 | 35 | 40 | 8.48 | .001 |
| L ventrolateral PFC | 44 | 354 | -48 | 26 | 34 | 9.52 | .001 |
| L temporoparietal junction | 40 | 483 | -45 | -46 | 52 | 9.22 | .001 |
| R inf parietal lobule | 40 | 141 | 39 | -43 | 43 | 7.33 | .001 |
| L sup parietal lobule | 7 | 282 | -24 | -64 | 55 | 9.92 | .001 |
| R sup parietal lobule | 7 | 190 | 27 | -61 | 43 | 12.08 | .001 |
| Precuneus | 7 | 243 | -9 | -76 | 49 | 9.33 | .001 |
| R thalamus | | 15 | 15 | -4 | 4 | 7.40 | .001 |
| L thalamus | | 100 | -15 | -10 | 1 | 8.72 | .001 |
| L hippocampus | 27 | 95 | -24 | -31 | 1 | 12.17 | .001 |
| R hippocampus | 37 | 43 | 24 | -28 | 1 | 14.29 | .001 |
| Occipital lobe/cuneus | 18 | 3010 | 18 | -94 | 10 | 19.50 | .001 |

*Note*: t(29) = 6.00, p < .05; FWE-corrected, k > 10

anticipation following cooperation, significant cluster-wise activity was detected in the right dmPFC, right TPJ, and right anterior insula (see Table 5 for results and details of thresholding).

During anticipation following defection, significant cluster-wise activity was detected more broadly across the prefrontal cortex and included the dmPFC, right dlPFC, and bilateral TPJ. However, anterior insula activity was not detected, mirroring what we observed in the decision contrasts (see Table 5).

**Feedback.** Across all four feedback conditions (reciprocated, unreciprocated, co-player cooperation, co-player defection), a common pattern of significant activation emerged that spanned the frontoparietal and salience networks of the brain. Active regions included the dmPFC, right dlPFC, bilateral vlPFC, the lateral OFC, the aMCC, the bilateral TPJ, bilateral

**Table 5. Significant regional activation during anticipation phases.**

| | | | MNI Coordinates | | | | |
|---|---|---|---|---|---|---|---|
| Name of Region | Brodmann Area | Voxels | x | y | z | t(29) | p-value (p < .001; Clusterwise-FDR corrected) |
| Anticipation (C) | | | | | | | |
| R dorsomedial PFC | 9 | 53 | 6 | 38 | 37 | 5.36 | .03 |
| R temporoparietal junction | 40 | 53 | 39 | -55 | 46 | 4.61 | .03 |
| R anterior insula | 47 | 91 | 36 | 26 | -5 | 5.51 | .01 |
| Occipital lobe/cuneus | 17 | 153 | 15 | -97 | 10 | 6.89 | .001 |
| Anticipation (D) | | | | | | | |
| Dorsomedial PFC | 8 | 155 | 9 | 35 | 46 | 5.55 | .001 |
| R dorsolateral PFC | 9 | 113 | 36 | 11 | 52 | 4.77 | .001 |
| L ventrolateral PFC | 45 | 56 | -45 | 47 | 1 | 4.87 | .01 |
| L temporoparietal junction | 40 | 217 | -39 | -58 | 43 | 5.05 | .001 |
| R temporoparietal junction | 40 | 232 | 60 | -31 | 49 | 4.70 | .001 |
| Occipital lobe, cuneus | 17 | 241 | 18 | -97 | 10 | 6.35 | .001 |

*Note*: All results were thresholded at $t(29) = 3.41$, $p < .001$ uncorrected voxel-wise threshold; cluster-wise FDR-corrected threshold determined by SPM12.

superior parietal lobules, the precuneus, the bilateral anterior insula, and the bilateral hippo-campi. Unique regions of activity included the right temporal pole during unreciprocated feedback and feedback following co-player defection and bilateral lateral OFC activity during reciprocated feedback and feedback following co-player cooperation. See Table 6 for detailed results.

## Human game direct contrasts analysis

Direct contrasts were delineated between sub-epochs within the three phases of the task (e.g., decision to cooperate versus decision to defect). The results of these analyses did not survive voxel-wise FWE correction, necessitating the use of a cluster-wise, FDR corrected threshold. (see Table 7). A direct contrast between the decision to cooperate versus the decision to defect revealed limited relative elevation of activation in the calcarine sulcus during the decision to cooperate, $t(29) = 5.05$, $p < .001$.

There was significantly greater activity in the hippocampus during anticipation following cooperation versus anticipation following defection, $t(29) = 4.96$, $p < .001$, while anticipation following defection elicited greater activity in the right dlPFC, $t(29) = 4.36$, $p < .001$, the pre-central gyrus, $t(29) = 5.55$, $p < .001$, and the postcentral gyrus, $t(29) = 4.82$, $p < .001$ (see Fig 4)

For the feedback conditions, a direct contrast between reciprocated and unreciprocated feedback revealed heightened activity in the precuneus when participants experienced unreci-procated feedback, $t(29) = 4.33$, $p < .001$. A similar result was found in the direct contrast between feedback following co-player cooperation and defection. Significant activity was elic-ited in the precuneus when participants experienced co-player defection in comparison to co-player cooperation, $t(29) = 5.05$, $p < .001$.

Lastly, direct contrasts were implemented between phases in the task irrespective of the decision made by the participant (see Table 8; see Fig 5). Significant activity was only detected in the Decision>Feedback, Decision>Anticipation, and Feedback>Anticipation contrasts. Within these contrasts, significant activity was identified in the dlPFC, vlPFC, TPJ, superior parietal lobules, anterior insula, bilateral hippocampi, and thalami. However, while the Deci-sion>Anticipation and Feedback>Anticipation contrasts also revealed activity in the right lat-eral OFC, aMCC, and precuneus, the Decision>Feedback contrast did not. Lastly, activity in

**Table 6. Significant regional activation during feedback phases.**

| Name of Region | Brodmann Area | Voxels | x | y | z | t(29) | p-value (p < .05; FWE-corrected) |
|---|---|---|---|---|---|---|---|
| | | | **MNI Coordinates** | | | | |
| **Reciprocated (CC+DD)** | | | | | | | |
| Dorsomedial PFC | 32 | 60 | 0 | 29 | 40 | 10.41 | .001 |
| R dorsolateral PFC | 9 | 660 | 39 | 29 | 43 | 8.92 | .001 |
| L ventrolateral PFC | 45 | 230 | -33 | 8 | 55 | 9.18 | .001 |
| R ventrolateral PFC | 45 | 133 | 45 | 44 | 25 | 9.83 | .001 |
| L lateral OFC | 11 | 42 | -45 | 50 | -2 | 8.55 | .001 |
| R lateral OFC | 11 | 189 | 39 | 50 | -11 | 7.55 | .001 |
| R ant midcingulate | 32 | 40 | 3 | 35 | 34 | 7.59 | .001 |
| L temporoparietal junction | 40 | 263 | -48 | -49 | 49 | 10.41 | .001 |
| R temporoparietal junction | 40 | 202 | 48 | -49 | 46 | 11.33 | .001 |
| R sup parietal lobule | 7 | 82 | 36 | -61 | 52 | 10.55 | .001 |
| L sup parietal lobule | 7 | 93 | -30 | -67 | 37 | 11.99 | .001 |
| Precuneus | 7 | 172 | 6 | -73 | 46 | 8.71 | .001 |
| R ant insula | 13 | 128 | 39 | 17 | -8 | 8.97 | .001 |
| L ant insula | 47 | 11 | -33 | 17 | -5 | 6.66 | .001 |
| R hippocampus | 37 | 22 | 24 | -28 | -2 | 7.60 | .001 |
| Occipital lobe/cuneus | 17 | 2645 | 15 | -94 | 7 | 12.71 | .001 |
| **FeeUnreciprocated (CD+DC)** | | | | | | | .001 |
| Dorsomedial PFC | 32 | 83 | 6 | 24 | 42 | 7.27 | .001 |
| R dorsolateral PFC | 9 | 109 | 48 | 17 | 43 | 8.49 | .001 |
| R ventrolateral PFC | 45 | 599 | 45 | 29 | 37 | 10.20 | .001 |
| L ventrolateral PFC | 45 | 330 | -45 | 26 | 31 | 7.96 | .001 |
| L lateral OFC | 46 | 47 | -42 | 50 | -2 | 6.96 | .001 |
| R ant midcingulate | 32 | 41 | 6 | 32 | 40 | 9.28 | .001 |
| R temporoparietal junction | 40 | 207 | 45 | -58 | 40 | 8.55 | .001 |
| L temporoparietal junction | 40 | 320 | -42 | -46 | 40 | 8.55 | .001 |
| R sup parietal lobule | 7 | 77 | 24 | -67 | 49 | 7.16 | .001 |
| L sup parietal lobule | 7 | 132 | -30 | -64 | 43 | 10.85 | .001 |
| Precuneus | 7 | 187 | 3 | -70 | 43 | 8.14 | .001 |
| R temporal pole | 38 | 152 | 42 | 20 | -20 | 9.20 | .001 |
| L ant insula | 47 | 102 | -30 | 17 | -17 | 9.67 | .001 |
| R ant insula | 47 | 73 | 30 | 17 | -14 | 9.06 | .001 |
| L hippocampus | 27 | 37 | -24 | -28 | -2 | 9.14 | .001 |
| R hippocampus | 37 | 43 | 24 | -28 | -2 | 9.52 | .001 |
| Occipital lobe/cuneus | 17 | 2329 | 18 | -94 | 7 | 10.92 | .001 |
| **FeeCo-Player Cooperation (CC+DC)** | | | | | | | |
| Dorsomedial PFC | 32 | 231 | 6 | 44 | 43 | 7.74 | .001 |
| R dorsolateral PFC | 9 | 145 | 39 | 11 | 52 | 10.03 | .001 |
| L ventrolateral PFC | 44 | 180 | -48 | 23 | 28 | 7.93 | .001 |
| R ventrolateral PFC | 48 | 69 | 51 | 32 | 28 | 9.07 | .001 |
| R lateral OFC | 46 | 63 | 36 | 53 | -2 | 7.14 | .001 |
| L lateral OFC | 47 | 31 | -42 | 50 | -5 | 8.74 | .001 |
| R ant midcingulate | 32 | 29 | 6 | 38 | 31 | 9.28 | .001 |
| R temporoparietal junction | 40 | 274 | 39 | -58 | 52 | 7.19 | .001 |
| L temporoparietal junction | 40 | 357 | -42 | -55 | 49 | 9.32 | .001 |
| R sup parietal lobule | 7 | 51 | 30 | -67 | 49 | 8.13 | .001 |

*(Continued)*

**Table 6.** (Continued)

| Name of Region | Brodmann Area | Voxels | MNI Coordinates | | | t(29) | p-value (p < .05; FWE-corrected) |
|---|---|---|---|---|---|---|---|
| | | | x | y | z | | |
| L sup parietal lobule | 7 | 51 | -27 | -67 | 49 | 7.34 | .001 |
| Precuneus | 7 | 57 | 6 | -73 | 40 | 7.88 | .001 |
| R hippocampus | 37 | 19 | 24 | -28 | -2 | 8.04 | .001 |
| L hippocampus | 27 | 13 | -24 | -31 | -2 | 7.47 | .001 |
| R ant insula | 48 | 58 | 36 | 20 | -8 | 7.56 | .001 |
| Occipital lobe | 17 | 2885 | 18 | -94 | 7 | 10.58 | .001 |
| FeeCo-Player Defection (CD+DD) | | | | | | | |
| Dorsomedial PFC | 8 | 154 | 3 | 26 | 43 | 10.82 | .001 |
| R dorsolateral PFC | 9 | 159 | 45 | 29 | 37 | 11.97 | .001 |
| L dorsolateral PFC | 46 | 59 | -39 | 23 | 40 | 10.81 | .001 |
| R ventrolateral PFC | 48 | 27 | 54 | 17 | 13 | 8.24 | .001 |
| L ventrolateral PFC | 45 | 115 | -45 | 29 | 31 | 10.26 | .001 |
| R lateral OFC | 11 | 69 | 30 | 47 | -14 | 7.96 | .001 |
| R ant midcingulate | 32 | 42 | 6 | 38 | 25 | 7.59 | .001 |
| L temporoparietal junction | 40 | 354 | -48 | -46 | 46 | 12.07 | .001 |
| R temporoparietal junction | 40 | 219 | 39 | -58 | 46 | 11.44 | .001 |
| L sup parietal lobule | 7 | 185 | -30 | -64 | 43 | 11.28 | .001 |
| R sup parietal lobule | 7 | 105 | 33 | -64 | 52 | 9.55 | .001 |
| Precuneus | 7 | 117 | -3 | -73 | 46 | 9.60 | .001 |
| R temporal pole | 38 | 133 | 45 | 17 | -17 | 7.52 | .001 |
| R ant insula | 48 | 90 | 30 | 17 | -11 | 10.26 | .001 |
| L ant insula | 47 | 93 | -30 | 17 | -14 | 8.98 | .001 |
| R hippocampus | 37 | 35 | 24 | -28 | -2 | 8.53 | .001 |
| L hippocampus | 27 | 20 | -21 | -28 | -5 | 7.43 | .001 |
| Occipital lobe | 17 | 2405 | 18 | -94 | 1 | 12.05 | .001 |

*Note*: $t(29) = 5.98$, $p < .05$ FWE-corrected, $k > 10$

the right temporal pole and posterior midcingulate was unique to the Feedback>Anticipation contrast. See S4 File and S1–S4 Tables for the results of the neuroimaging analysis for the computer game. See S5 File and S5–S9 Tables for neuroimaging findings directly contrasting neural activity between human and computer gameplay.

## Discussion

The goal of the current study was to characterize neural substrates of reciprocal social exchange, modelled as phases of a prospective "social decision cascade" using the iPD task. As expected, we found both common and distinct patterns of neural activity across phases. Our results reaffirm findings in the iPD literature that implicate functional nodes associated with social reasoning as active during decision-making and feedback appraisal [2–6]. Additionally, our findings offer tentative support for the significance of internal conflict as a predictor of regional activity during the anticipation phase. Notably, only structures previously implicated in ToM reasoning (TPJ and dmPFC; [39, 40] reached a significant threshold of activation during all phases of the task. Our results, taken collectively, paint a more complete picture of the social decision cascade and its neural correlates than the literature to date has presented, and several key points warrant mention.

**Table 7. Regional activation during direct comparison contrasts (within-phase).**

| Name of Region | Brodmann Area | Voxels | MNI Coordinates x | y | z | t(29) | p-value (p < .001; Clusterwise-FDR corrected) |
|---|---|---|---|---|---|---|---|
| Decision (C) > Decision (D) | | | | | | | |
| L Calcarine Sulcus | 18 | 171 | -15 | -76 | 8 | 5.05 | .001 |
| Decision (D) > Decision (C) | | | | | | | |
| No suprathreshold voxels | | | | | | | |
| Anticipation (C) > Anticipation (D) | | | | | | | |
| R hippocampus | 27 | 41 | 39 | -28 | -8 | 4.96 | .05 |
| Anticipation (D) > Anticipation (C) | | | | | | | |
| R precentral gyrus | 4 | 510 | 33 | -22 | 55 | 5.55 | .001 |
| L postcentral gyrus | 2 | 229 | -18 | -37 | 70 | 4.82 | .001 |
| R dorsolateral PFC | 9 | 69 | 27 | 29 | 52 | 4.36 | .01 |
| Reciprocated > Unreciprocated | | | | | | | |
| No suprathreshold voxels | | | | | | | |
| Unreciprocated > Reciprocated | | | | | | | |
| R precuneus | 7 | 100 | 3 | -58 | 43 | 4.33 | .01 |
| Co-Player Cooperation > Defection | | | | | | | |
| No suprathreshold voxels | | | | | | | |
| Co-Player Defection > Cooperation | | | | | | | |
| L precuneus | 7 | 403 | -21 | -61 | 64 | 5.05 | .001 |

*Note*: t(29) = 3.38, p < .001 uncorrected voxel-wise threshold; FWE-corrected cluster-wise threshold determined by SPM12.

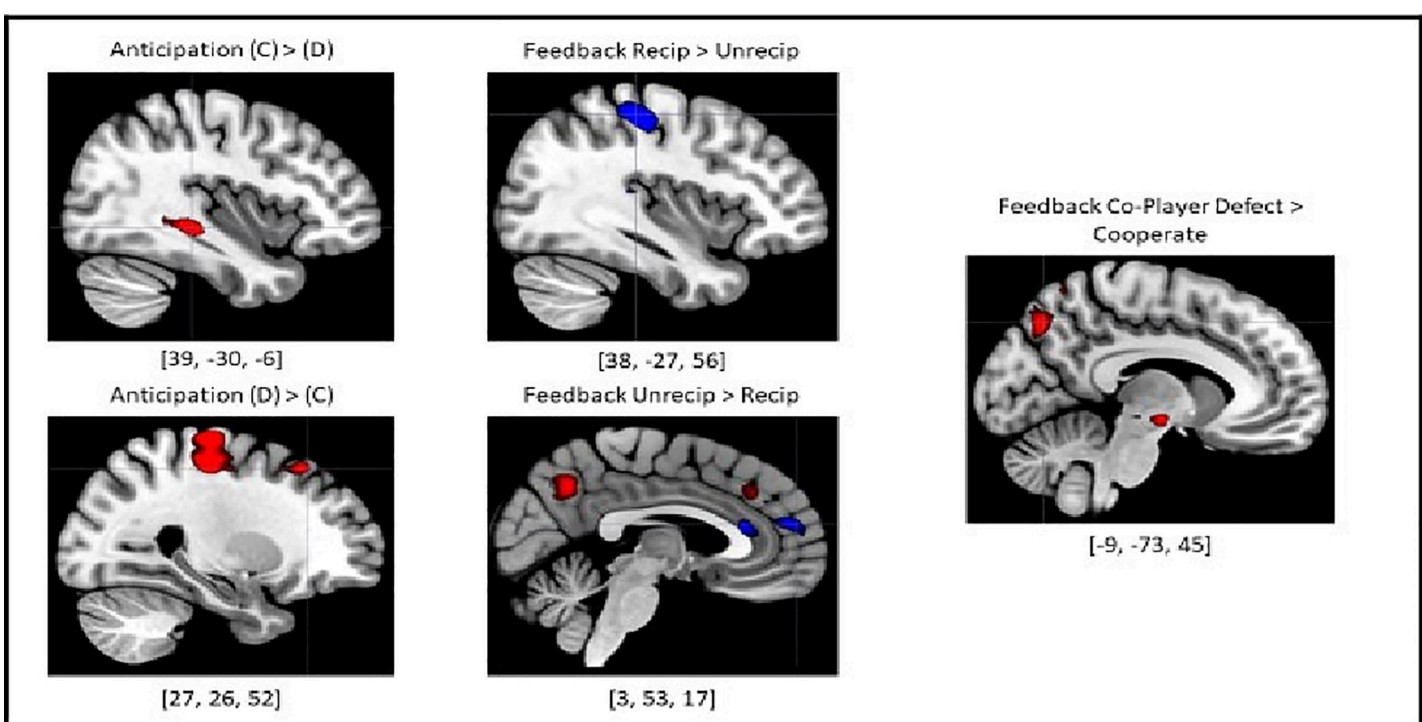

**Fig 4. BOLD activity illustrating direct comparison contrasts within-phase.** The most salient finding was the activation of the precuneus, involved in self-referential processing, during aversive social outcomes regardless of context (monetary vs social). This result also supports the precuneus's potential role in conflict monitoring and social adaptation in response to negative outcomes. All results for these contrasts were thresholded at t(29) = 3.40, p < .001 uncorrected voxel-wise threshold; FWE-corrected cluster-wise threshold determined by SPM12. (**Red = Human Game, Blue = Computer Game, C = Cooperate, D = Defect**). Ex. [42, –48, 42] = peak voxel MNI coordinates.

**Table 8. Regional activation during direct comparison contrasts (between-phase).**

| Name of Region | Brodmann Area | Voxels | x | y | z | t(29) | p-value (p < .05; FWE-corrected) |
|---|---|---|---|---|---|---|---|
| | | | **MNI Coordinates** | | | | |
| Decision > Feedback | | | | | | | |
| R dorsolateral PFC | 9 | 44 | 27 | 53 | 34 | 8.29 | .001 |
| L ventrolateral PFC | 45 | 62 | -45 | 41 | 22 | 7.86 | .001 |
| L temporoparietal junction | 40 | 419 | -42 | -37 | 43 | 11.21 | .001 |
| R inf parietal lobule | 40 | 64 | 30 | -43 | 43 | 10.28 | .001 |
| L sup parietal lobule | 7 | 231 | -18 | -67 | 46 | 8.61 | .001 |
| R sup parietal lobule | 7 | 143 | 15 | -67 | 58 | 8.39 | .001 |
| L ant insula | 13 | 15 | -48 | 14 | -5 | 6.75 | .01 |
| R hippocampus | 27 | 31 | 24 | -31 | 1 | 11.57 | .001 |
| L hippocampus | 37 | 33 | -18 | -31 | -2 | 10.70 | .001 |
| L thalamus | | 173 | -15 | -10 | 1 | 8.90 | .001 |
| R thalamus | | 70 | 15 | -7 | 1 | 8.13 | .001 |
| L precentral gyrus | 6 | 51 | -48 | 2 | 28 | 8.22 | .001 |
| R cerebellum 8 | | 64 | 15 | -64 | -44 | 8.09 | .001 |
| Mid occipital lobe | 17 | 2956 | 30 | -91 | 4 | 16.27 | .001 |
| Feedback > Decision | | | | | | | |
| No suprathreshold voxels | | | | | | | |
| Decision > Anticipation | | | | | | | .001 |
| R dorsolateral PFC | 9 | 555 | 27 | 53 | 34 | 12.08 | .001 |
| L dorsolateral PFC | 46 | 194 | -33 | 56 | 19 | 8.77 | .001 |
| L ventrolateral PFC | 45 | 367 | -45 | 29 | 34 | 10.74 | .001 |
| R lateral OFC | 11 | 30 | 30 | 56 | -11 | 7.35 | .001 |
| Ant midcingulate | 32 | 94 | 9 | 29 | 28 | 8.36 | .001 |
| R post midcingulate | 23 | 71 | 0 | -25 | 28 | 9.40 | .001 |
| L temporoparietal junction | 40 | 549 | -36 | -46 | 40 | 14.22 | .001 |
| R temporoparietal junction | 40 | 234 | 39 | -40 | 40 | 9.94 | .001 |
| L sup parietal lobule | 7 | 302 | -24 | -64 | 52 | 9.39 | .001 |
| R sup parietal lobule | 7 | 221 | 36 | -58 | 55 | 9.42 | .001 |
| Precuneus | 7 | 351 | 12 | -70 | 49 | 10.45 | .001 |
| L ant insula | 47 | 52 | -39 | 14 | -2 | 7.09 | .001 |
| L hippocampus | 27 | 40 | -24 | -34 | -2 | 13.08 | .001 |
| R hippocampus | 37 | 39 | 24 | -28 | -2 | 14.18 | .001 |
| L thalamus | | 195 | -12 | -16 | 7 | 8.97 | .001 |
| R thalamus | | 110 | 15 | -13 | 10 | 7.53 | .001 |
| Occipital lobe, cuneus | 18 | 3148 | 18 | -94 | 10 | 15.77 | .001 |
| Anticipation > Decision | | | | | | | |
| No suprathreshold voxels | | | | | | | |
| Feedback > Anticipation | | | | | | | |
| R lateral OFC | 111 | 119 | 24 | 50 | -14 | 7.88 | .001 |
| R dorsolateral PFC | 9 | 1065 | 30 | 29 | 49 | 9.23 | .001 |
| L dorsolateral PFC | 46 | 111 | -42 | 50 | 7 | 10.09 | .001 |
| R ventrolateral PFC | 45 | 111 | 51 | 29 | 28 | 10.80 | .001 |
| L ventrolateral PFC | 45 | 368 | -42 | 29 | 31 | 11.51 | .001 |
| R ant midcingulate | 32 | 82 | 6 | 35 | 25 | 11.13 | .001 |
| R post midcingulate | 23 | 71 | 3 | -25 | 31 | 7.84 | .001 |
| L temporoparietal junction | 40 | 468 | -42 | -49 | 43 | 13.74 | .001 |

*(Continued)*

**Table 8.** (Continued)

| Name of Region | Brodmann Area | Voxels | MNI Coordinates | | | t(29) | p-value (p < .05; FWE-corrected) |
| --- | --- | --- | --- | --- | --- | --- | --- |
| | | | x | y | z | | |
| R temporoparietal junction | 40 | 266 | 42 | -49 | 40 | 12.22 | .001 |
| L sup parietal lobule | 7 | 215 | -30 | -64 | 43 | 10.62 | .001 |
| R sup parietal lobule | 7 | 133 | 33 | -67 | 49 | 10.47 | .001 |
| Precuneus | 7 | 432 | 9 | -70 | 40 | 10.83 | .001 |
| R temporal pole | 38 | 145 | 45 | 20 | -17 | 10.56 | .001 |
| R ant insula | 47 | 63 | 36 | 17 | -14 | 7.62 | .01 |
| L ant insula | 47 | 23 | -45 | 17 | -8 | 7.03 | .01 |
| R hippocampus | 37 | 131 | 24 | -28 | -5 | 12.17 | .001 |
| L hippocampus | 27 | 87 | -31 | -31 | 1 | 9.70 | .001 |
| L mid occipital lobe | | 507 | -36 | -88 | -5 | 10.44 | .001 |
| R mid occipital lobe | | 353 | 33 | -79 | 10 | 10.02 | .001 |
| L cerebellum 9 | | 56 | -6 | -55 | -50 | 8.10 | .001 |
| Anticipation > Feedback | | | | | | | |
| No suprathreshold voxels | | | | | | | |

*Note*: t(29) = 6.0, p < .05 FWE-corrected voxel-wise threshold, k > 10

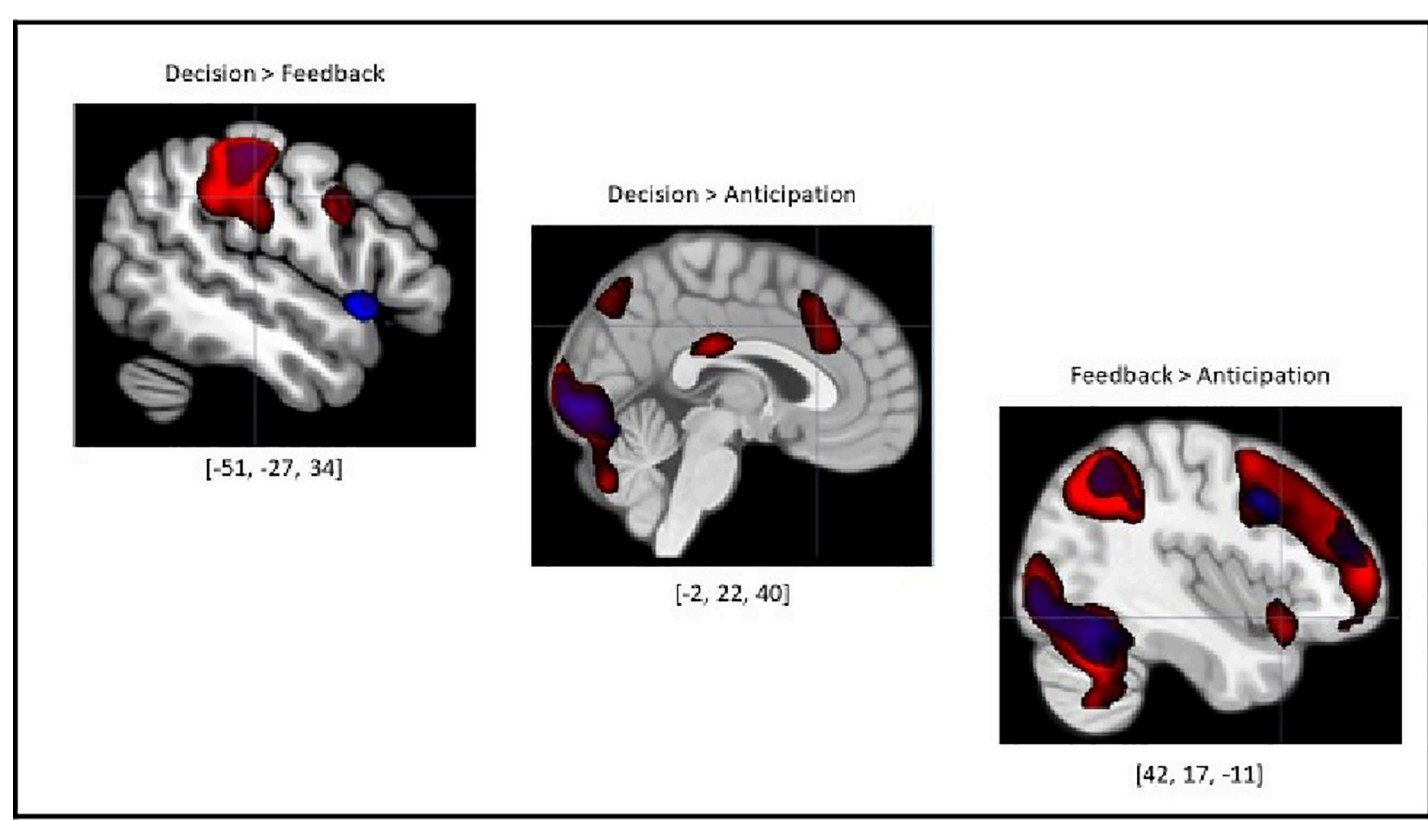

**Fig 5. BOLD activity displaying direct comparison contrasts between phases.** The decision-making phase exhibited the strongest neural signature in comparison to the other phases of the task, suggesting a significantly elevated cognitive and neural resource requirement for that phase. All contrasts were thresholded at t(29) = 6.00, p < .05; FWE-corrected voxel-wise threshold. Direct contrasts not displayed returned no suprathreshold voxels of activation using voxel-wise or cluster-wise thresholding. (**Red = Human Game, Blue = Computer Game, C = Cooperate, D = Defect**). Ex. [42, −48, 42] = peak voxel MNI coordinates.

We had hypothesized that activity in the dmPFC, the caudate, rostral ACC/aMCC, and the TPJ would significantly be associated with decision-making in the task. Our results revealed a larger distribution of regions that, taken together, could constitute a social decision-making network. This network included among its nodes the dmPFC, aMCC and the left TPJ; however, activity in the caudate was absent. This finding was unexpected especially given the wealth of literature that supports the caudate's involvement in mediating goal-oriented decision-making and adaptive behavior in social tasks [41–43]. A possible explanation is that the repetitive nature of the task diminished the motivational efficacy of the reward derived from gameplay and rendered goal formation inconsequential; however this outcome is unlikely given previous iPD findings [3, 44, 45]. A more plausible explanation could be that our preprocessing pipeline, which included smoothing images with a relatively large Gaussian kernel of 8mm FWHM as a noise reduction approach, may have hindered detection of localized brain activity within small, subcortical structures such as the striatum and the amygdala [46]. A number of additional regions that had been implicated in iPD studies targeting specific populations were recruited; these included the precuneus and the anterior insula. Of note, we also observed activity in the dmPFC and the vlPFC, which are regions that had not shown significant activation in prior iPD studies. These findings suggest additional regions to consider as salient to social decision-making in the context of the iPD.

Of all of the phases, the feedback phase has been the most clearly characterized in the iPD fMRI literature to date [3–6, 13]. The present findings extend this work by providing evidence of significant engagement during feedback of our hypothesized social decision-making network, along with the dlPFC, vlPFC, precuneus, and temporal pole. Previous iPD studies had detected no more than three of these regions activated concurrently during trials; our findings, however, suggest the possibility of large-scale interdependency among these neuroanatomical structures that supports the appraisal of either social harmony/conflict or monetary gain/loss. This interpretation is bolstered by the observation that the same regions were recruited across each feedback outcome (Social outcome: reciprocated and unreciprocated; Monetary outcome: co-player cooperation and co-player defection). Additionally, the majority of the detected regions are part of the frontoparietotemporal network of the brain, a multi-faceted and dynamic network that mediates cognitive flexibility, problem-solving and emotional regulation more broadly in task-related contexts [47, 48]. This observation provides a foundation for future functional connectivity analyses that could elucidate whether the neural substrates of social reasoning conform to a larger coherent executive functioning network or whether social cognition and executive functioning are dissociable at the neural level [49–53].

A qualitative examination of findings for the decision and feedback phases of iPD rounds revealed that they share substantially overlapping neural networks. We observed common activation across these phases in several regions involved in subjective valuation, social reasoning and problem solving such as the lateral frontal gyri, the dmPFC, lateral OFC, bilateral TPJ, precuneus, bilateral anterior insula, bilateral hippocampi and the bilateral thalami [14, 40, 54]. The striking similarities in patterns of activation during the decision and feedback phases raise the possibility that the same neural networks contribute to both, at a domain-general level. However, it is difficult to say, based on our data, whether these networks would support social reasoning specifically or general executive functioning more broadly.

However, a direct contrast between the two phases yielded evidence that the phases might also appropriately be treated as distinct. In particular, all commonly activated neural regions exhibited significantly stronger BOLD activity during decision-making than during feedback. This finding suggests that the mental processes involved in making a social choice and those involved in processing feedback about whether one's social choice yields reward or punishment draw on a common neural network, but do so in different ways. Stronger inferences

concerning the behavior of these networks could be facilitated through the utilization of robust functional connectivity techniques. For example, mapping changes in dynamic functional connectivity within the network over the course of task engagement could be an appropriate approach to determining the magnitude and temporal stability of network recruitment over the course of gameplay. Network dynamics research suggests that moment-to-moment fluctuations in functional connectivity are more stable during complex task performance than at rest and that interregional neuronal hubs often reorganize during tasks, a phenomenon that could easily be captured with the iPD, given the breadth of regional activation identified in this study [55]. However, such research would require a much larger participant sample than we recruited for the present study.

Another direct contrast within the decision-making phase of the game revealed an amplified neural response in the visual cortex when making cooperative versus defection-based decisions. PD fMRI literature overwhelmingly suggests that most individuals oftentimes find the prospect of defecting against their partner to be more aversive and conflict-laden that cooperating [3, 4, 13], which we posited would evoke a heightened neural signature during defection in a direct contrast to cooperation. This outcome may possibly be the result of our reduced dilemma strength for our version of the PDG ($D_g' = D_r' = 1$). It is possible that the various outcomes did not register as significantly differentiated for our participants, which could also explain the lack of reward-based striatal activity over the course of the task. If the dilemma strength was increased by a factor of 5, for example, to increase the earning stakes of the game, this could significantly augment the neural response to defection by intensifying the desire to betray the partner for elevated gains, even while risking periods of social disintegration due to fear of betrayal from the participant. This is another avenue that we believe is worth exploration in future research utilizing this paradigm., especially given the fact that the vast majority of PD fMRI research also employs this same reduced strength variant as the prototypical model of the PD.

Although our findings regarding activity during the anticipation phase are necessarily tentative, given that significant activations only emerged at a liberal cluster-wise threshold, they offer potential evidence that anticipation following participant defection is functionally dissociable from anticipation following participant cooperation. This finding raises the possibility that sensitivity to social conflict may modulate brain activity during the anticipatory process. Anticipation following the decision to cooperate elicited a cluster of activity within the anterior insula, while anticipation following defection elicited clusters of activity within the aMCC, right dlPFC, dmPFC, and bilateral TPJ. Striatum activity was absent in this phase, remaining consistent with observations in the decision and feedback phases.

With regard to anticipation following cooperation, research suggests that the anterior insula is involved in self-awareness and subjective emotional experience [56, 57]. In the context of social interaction, this region has been implicated in mediating affective empathic response, and more specifically generating shared representations of the feelings others [58, 59]. It is possible that the anterior insula facilitated reciprocation in the iPD task by processing emotions associated with desirable positive outcomes of decisions [60]. If this interpretation is accurate, it would align with prior evidence that humans receive emotional reward from cooperating with their peers and anticipating positive outcomes in the social situations such as the PD [61, 62]. However, the anterior insula has also been heavily implicated in associative fear-based learning and the anticipation and processing of aversive outcomes [56, 63, 64]. From this viewpoint, engagement of the anterior insula may signal a fear of betrayal after making the risky decision to cooperate with an unpredictable and previously unknown social partner. Greater insight into the motivations and perceptions of the participants is required to decompose these possibilities.

In contrast, anticipation following defection elicited activity in a network involving aMCC, dmPFC, right dlPFC, and TPJ. Converging neural activity within the rostral ACC and dlPFC has been associated with conflict monitoring/resolution, cognitive control and feedback-mediated decision-making based on the evaluation of previous action outcomes [65–67]. Furthermore, the joint recruitment of the TPJ and dmPFC could suggest activation of a network that differentiates the processing of socio-cognitive conflict (e.g., social cues suggesting conflicting assumptions about the subjective states of an individual) from the processing of general conflict (e.g., interpreting descriptive or declarative sentences) [68] It is possible that participants perceive the decision to defect as conflict-laden because it contradicts social norms, and thus must recruit greater cognitive and neural resources to prepare for the prospect of further social conflict introduced by the participant or the co-player. Alternatively, these regions could be operating independently and be constrained by domain-specific functions attached to the anticipatory process. Replication in larger datasets will be necessary to obtain more robust results and examine each of these possibilities; the current results, however, provide a starting point from which to approach the examination of this phase of the social decision cascade.

We found evidence that the TPJ and the dmPFC were both active during all phases of the cascade. This finding merits note, given that multiple meta-analytic studies have identified these areas as "core" nodal regions underlying social cognition and Theory of Mind (ToM). ToM is defined as the ability to attribute goals, intentions, and beliefs to other individuals [69–71]. The TPJ has specifically been implicated in third person perspective-taking and in creating temporary representations of other peoples' mental states [72, 73], while the dmPFC appears to support inferences from a first person perspective about stable personality dispositions of the self as well as others, along with mediating the application of social norms and scripts [71]. These functions are essential during a task like the iPD game, which requires participants to attempt to predict partner behavior on a round by round basis. They are asked to think from their own and their partner's perspectives and coordinate responses with the partner in order to mutually benefit from the interaction or adapt to the partner's inconsistent behavior [74]. The iPD task should thus tend to consistently recruit social cognition nodes such as the TPJ and dmPFC, and the presence of activation in these regions could serve as a quality check when assessing the effectiveness and utility of the iPD task as a model of social interaction.

Lastly, the anterior insula showed stronger activation when processing "human" co-player defection versus computer co-player defection, based on findings from a direct comparison of brain activity following these negative outcomes. Previous iPD findings support the idea that people tend to treat their decisions as more important and to show more willingness to conform to social standards when they believe their partner is a human being whose actions are perceived as more deliberate and intentional [75, 76]. It is unsurprising then that co-player defection was met with a salient aversive response in the anterior insula when perpetrated by a human partner rather than a computer partner.

A few limitations of this study warrant mention. First, data were collected at two independent sites, with several years separating time of collection. Scanner and MR sequence protocols diverged across the two studies, which necessitated data correction during preprocessing to ensure that parameters were consistent across the final dataset (see S4 Fig in the supplemental materials for a comparison of activity between our two datasets). We included site as a regressor in all analyses to mitigate the impact of site effects. Second, a number of participants defected much more often than they cooperated. Consequently, only a limited number of CD trials could be sampled for those participants. These contrast images were still included in the subsequent group analysis, which could have reduced the overall power of the analysis. Another limitation was the lack of an explicit baseline condition (e.g., cross fixation). All periods in which the participant was not engaged in playing the PD task or answering emotional

assessment questions acted as an "implicit" baseline. An additional limitation concerning the emotional assessment questions is the possibility that interruptions during gameplay could have affected the "natural" experience and engagement during play. We decided to use an "online" rating with the knowledge that while disrupting the flow of an ongoing task might alter the emotions being measured, online ratings do show a resistance to degeneration due to memory limitations and allow researcher to more effectively probe emotions related to specific events [25].

Our findings complement the existing iPD fMRI literature by providing evidence that a shared neural network may support social cognition during decision-making and feedback appraisal. The findings lend support to the idea that the dmPFC and TPJ play roles in social reasoning and suggest that these regions should be focal points of analysis in future economic-exchange studies. We also provide a foundation for the examination of the anticipation phase of iPD task rounds, which has received little research attention to date. Future research aimed at replicating and extending our findings regarding the social decision cascade could help further validate the use of economic exchange paradigms as models of social interaction in social neuroscience research.

## Supporting information

**S1 File. Elaboration of the deception protocol.**
(DOCX)

**S2 File. Description of computerized co-player algorithm.**
(DOCX)

**S3 File. Description of jittered ISI in task design.**
(DOCX)

**S4 File. Computer neuroimaging analysis.**
(DOCX)

**S5 File. Neuroimaging analysis of the first human PD game vs. computer game.**
(DOCX)

**S1 Fig. Time course of one Prisoner's Dilemma game.** The chart progresses from left-to-right, top-to-bottom. The first section details the timing of one round. The round always begins at "show round", "Get Ready" only preceded the first round. The second section details the timing of one block, which includes 5 trials and a set of four assessment questions. The final section details the timing of the entire four block run which concludes with 11 debriefing questions to end the game.
(TIF)

**S2 Fig. Design matrix of the 18-condition task.**
(TIF)

**S3 Fig. Comparison of BOLD activity between the two datasets within the feedback phase of the task.** There were no significant differences in patterns of activation between the datasets. <u>Decision:</u> (GSU, $t(16) = 3.69$, $p < .001$ uncorrected; Emory, $t(13) = 3.85$, $p < .001$ uncorrected). <u>Anticipation:</u> (GSU, $t(16) = 1.80$, $p < .05$ uncorrected; Emory, $t(13) = 1.75$, $p < .05$ uncorrected). <u>Feedback:</u> (GSU, $t(16) = 3.89$, $p < .001$ uncorrected; Emory, $t(13) = 3.85$, $p < .001$ uncorrected).
(TIF)

**S4 Fig. The left anterior insula (AI) exhibited greater BOLD response to unreciprocated cooperation (CD) when experienced during human play (1st game) in contrast to computer play [$t(29)$ = 3.43, p < .001 voxel-wise uncorrected, p < .05 FWE-cluster correction, k = 44, peak voxel = -30, 14, -11].**
(TIF)

**S1 Table. Significant regional activation during decision phases with computer.**
(DOCX)

**S2 Table. Significant regional activation during anticipation phases with computer.**
(DOCX)

**S3 Table. Significant regional activation during feedback phases with computer.**
(DOCX)

**S4 Table. Regional activation during direct comparison contrasts with computer.**
(DOCX)

**S5 Table. Significant peak voxels during the decision-making phase of the first PD game.**
(DOCX)

**S6 Table. Significant peak clusters during the anticipation phase of the first PD game.**
(DOCX)

**S7 Table. Significant peak voxels during the feedback phase of the first PD game.**
(DOCX)

**S8 Table. Direct contrasts within and between phases for the first PD game.**
(DOCX)

**S9 Table. Direct comparison between human (1st game) and computer BOLD activity during PD gameplay.**
(DOCX)

**S1 Material.**
(SAV)

## Author Contributions

**Conceptualization:** Eddy Nahmias, Negar Fani, Trevor Kvaran.

**Data curation:** Khalil Thompson, Eddy Nahmias, Trevor Kvaran.

**Formal analysis:** Khalil Thompson.

**Funding acquisition:** Khalil Thompson, Eddy Nahmias, Negar Fani, Jessica Turner, Erin Tone.

**Investigation:** Khalil Thompson, Eddy Nahmias, Negar Fani, Jessica Turner, Erin Tone.

**Methodology:** Khalil Thompson, Eddy Nahmias, Negar Fani, Trevor Kvaran, Jessica Turner, Erin Tone.

**Project administration:** Khalil Thompson, Eddy Nahmias, Negar Fani, Trevor Kvaran, Jessica Turner, Erin Tone.

**Resources:** Eddy Nahmias, Jessica Turner, Erin Tone.

**Software:** Jessica Turner, Erin Tone.

**Supervision:** Khalil Thompson, Eddy Nahmias, Negar Fani, Trevor Kvaran, Jessica Turner, Erin Tone.

**Validation:** Khalil Thompson, Negar Fani, Trevor Kvaran, Jessica Turner, Erin Tone.

**Visualization:** Khalil Thompson.

**Writing – original draft:** Khalil Thompson.

**Writing – review & editing:** Khalil Thompson.

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
