## [Decision Letter · Decision Letter 0]

18 Jan 2021

PONE-D-20-41107

The Prisoner’s Dilemma paradigm provides a neurobiological framework for the social decision cascade

PLOS ONE

Dear Dr. Thompson,

Thank you for submitting your manuscript to PLOS ONE. After careful consideration, we feel that it has merit but does not fully meet PLOS ONE’s publication criteria as it currently stands. Therefore, we invite you to submit a revised version of the manuscript that addresses the points raised during the review process.

We look forward to receiving your revised manuscript.

Kind regards,

Jun Tanimoto

Academic Editor

PLOS ONE

Journal Requirements:

2. Please note that according to our submission guidelines (http://journals.plos.org/plosone/s/submission-guidelines), outmoded terms and potentially stigmatizing labels should be changed to more current, acceptable terminology. For example: “Caucasian” should be changed to “white” or “of [Western] European descent” (as appropriate).

3. Please improve statistical reporting and refer to p-values as "p<.001" instead of "p=.000". Our statistical reporting guidelines are available at https://journals.plos.org/plosone/s/submission-guidelines#loc-statistical-reporting

Reviewers' comments:

Reviewer's Responses to Questions

**Comments to the Author**

1. Is the manuscript technically sound, and do the data support the conclusions?

Reviewer #1: Yes

Reviewer #2: Yes

2. Has the statistical analysis been performed appropriately and rigorously? 

Reviewer #1: Yes

Reviewer #2: Yes

3. Have the authors made all data underlying the findings in their manuscript fully available?

Reviewer #1: Yes

Reviewer #2: Yes

4. Is the manuscript presented in an intelligible fashion and written in standard English?

Reviewer #1: Yes

Reviewer #2: Yes

5. Review Comments to the Author

Reviewer #1: This work reports on an interesting experimental trail that fMRI from subjects who exposed to iPD games was systematically obtained so as to unfold the neural signature in relation with human-decision making process for a social interaction. Although the reported result seems still staying at primitive, it could be seen a good step as the inception aiming the goal as above.

One of the findings they reported struck me as interesting is that there is less significant evidence to distinct the neural signal entailed with defection as a subject’s decision from that when he drawing cooperation as his decision.

As a whole I can embrace a positive feeling on the MS. Yet, I would like to give the authors following suggestions to improve their MS.

#1.

This is a quite technical but crucially important. The authors obeyed to confused depiction when presenting so-called payoff matrix. That is the presentation in Fig. 1. I do believe that the row and column are inversely presented. If obeying to the standard notation, what they described in Fig. 1 means; R=$2, T=0, S=3 and P=1. Let alone this is not PD but Trivial game. I do believe that they imposed as PD was; R=2, =3, S=0 and P=1. They should fix it.

#2.

Again, their PD game is; R=2, =3, S=0 and P=1. Referring to the concept of universal dilemma strength by following works (the authors should cite in the revised MS);

Tanimoto & Sagara; Relationship between dilemma occurrence and the existence of a weakly dominant strategy in a two-player symmetric game, BioSystems 90(1), 105-114, 2007.

Wang et al.; Universal scaling for the dilemma strength in evolutionary games, Physics of Life Reviews 14, 1-30, 2015.

Ito et al.; Scaling the phase- planes of social dilemma strengths shows game-class changes in the five rules governing the evolution of cooperation, Royal Society Open Science, 181085, 2018.

Arefin et al.; Social efficiency deficit deciphers social dilemmas, Scientific Reports 10, 16092, 2020.

Their game has; Chicken-type dilemma; Dg’ = (T – R) / (R – P) =1 and Stag Hunt-type dilemma; Dr’ = (P – S) / (R – P) =1, which belongs to what-is-called Donor & Recipient (D & R) game. D & R game (concerning D & R game, they should reference; Evolutionary Games with Sociophysics: Analysis of Traffic Flow and Epidemics, Springer, 2019.)has been commonly applied especially by theoretical biologists as the standardized template for PD games, since both Dg’ and Dr’ exist but the game can be parameterized by the single dilemma parameter; Dg’=Dr’.

Incidentally, one of the authors’ findings was that when a subject drawing the decision of D and C, a neural signal seems less distinctive. One reason for this I guess is that their PD game has relatively less dilemma strength. Thus, I wonder there might be bit different results if they compare with the result from a much more severe dilemma situation, say; for instance; Dg’ =Dr’ =5. I wouldn’t go so far as say that further evidence should be obtained. But I suggest them to give further discussion on this point and mention on their future work relating to this point.

Reviewer #2: Based on iterated prisoner’s dilemma game, the authors used the method of functional magnetic resonance imaging (fMRI) to studied the neural activity associated with the three phases of the cascade during the social interactions. They presented some super interesting results and it reminds me of experimental work by Stuart A. West (Prosocial preferences do not explain human cooperation in public-goods games). In West’s paper, they conclude that prosocial preferences do not explain human cooperation by comparing the results in standard public goods game and the results in black-box game. Different with West’s paper, the authors analyzed and compared the neural activity between the ‘human’ and computer games. The manuscript is well-written, the methods are reasonable, and the statistical analysis are performed appropriately. I am happy to recommend it for publication if the following issues are addressed.

1. Figure captions are too short, in order make it more readable, please add one or two sentences to conclude its main conclusions. Please note that not all the readers have patients to read the whole text, add this would help the reader immensely.

2. Page 11, line183, and page 13, line 230-236 rhs. There are 40 rounds in human games and 20 rounds in computer games, I am not clear why the authors organized three treatments rather than four? Please note that when you calculate the cooperation rate in human and computer games, the baseline is different. Please clarify and expand.

3. Although the present paper is greatly different with West’s paper, I think there are some connections. In West’s paper, participants are unaware of its opponent’s information although they play with human, they just know that they input one number and will get a reward. In your paper, when participants play in ‘human’ games, the information of its opponent’s was almost complete, but the participants were wrongly thinking its opponents are human. If possible, please consider the connections and expand.

6. PLOS authors have the option to publish the peer review history of their article (what does this mean?). If published, this will include your full peer review and any attached files.

Reviewer #1: No

Reviewer #2: No

---

## [Author Response · Author response to Decision Letter 0]

7 Feb 2021

Academic Editor

The manuscript has been edited to adhere to the formatting guidelines of PLOS ONE. Comments have been attached to the marked up copy of the manuscript pointing to these edits.

2) Please note that according to our submission guidelines (http://journals.plos.org/plosone/s/submission-guidelines), outmoded terms and potentially stigmatizing labels should be changed to more current, acceptable terminology. For example: “Caucasian” should be changed to “white” or “of [Western] European descent” (as appropriate).

Changes have been made to endorse the use of contemporary terminology for various races and ethnicities in the manuscript.

3) Please improve statistical reporting and refer to p-values as "p<.001" instead of "p=.000". Our statistical reporting guidelines are available at https://journals.plos.org/plosone/s/submission-guidelines#loc-statistical-reporting

Statistics table have now been edited so that significance thresholds are estimated to be less than .001 and not equal to .000.

4) Please include captions for your Supporting Information files at the end of your manuscript, and update any in-text citations to match accordingly. Please see our Supporting Information guidelines for more information: http://journals.plos.org/plosone/s/supporting-information

Captions for Supporting files have been inserted at the end of the manuscript and in-text citations have been edited to match the new labeling conventions.

Reviewer #1 

1) This is a quite technical but crucially important. The authors obeyed to confused depiction when presenting so-called payoff matrix. That is the presentation in Fig. 1. I do believe that the row and column are inversely presented. If obeying to the standard notation, what they described in Fig. 1 means; R=$2, T=0, S=3 and P=1. Let alone this is not PD but Trivial game. I do believe that they imposed as PD was; R=2, =3, S=0 and P=1. They should fix it.

We profusely apologize for the confusion caused by the mislabeling of the payoff matrix in the illustration provided for Figure 1. The payoff matrix as described by Reviewer 1 was presented to participants during data collection, the inaccurate labelling as presented beforehand was simply an error in memory regarding its illustration during the designing of the figure. This mistake has now been fixed.

2) Again, their PD game is; R=2, =3, S=0 and P=1. Referring to the concept of universal dilemma strength by following works (the authors should cite in the revised MS);

Tanimoto & Sagara; Relationship between dilemma occurrence and the existence of a weakly dominant strategy in a two-player symmetric game, BioSystems 90(1), 105-114, 2007.

Wang et al.; Universal scaling for the dilemma strength in evolutionary games, Physics of Life Reviews 14, 1-30, 2015.

Ito et al.; Scaling the phase- planes of social dilemma strengths shows game-class changes in the five rules governing the evolution of cooperation, Royal Society Open Science, 181085, 2018.

Arefin et al.; Social efficiency deficit deciphers social dilemmas, Scientific Reports 10, 16092, 2020.

Their game has; Chicken-type dilemma; Dg’ = (T – R) / (R – P) =1 and Stag Hunt-type dilemma; Dr’ = (P – S) / (R – P) =1, which belongs to what-is-called Donor & Recipient (D & R) game. D & R game (concerning D & R game, they should reference; Evolutionary Games with Sociophysics: Analysis of Traffic Flow and Epidemics, Springer, 2019.)has been commonly applied especially by theoretical biologists as the standardized template for PD games, since both Dg’ and Dr’ exist but the game can be parameterized by the single dilemma parameter; Dg’=Dr’.

Incidentally, one of the authors’ findings was that when a subject drawing the decision of D and C, a neural signal seems less distinctive. One reason for this I guess is that their PD game has relatively less dilemma strength. Thus, I wonder there might be bit different results if they compare with the result from a much more severe dilemma situation, say; for instance; Dg’ =Dr’ =5. I wouldn’t go so far as say that further evidence should be obtained. But I suggest them to give further discussion on this point and mention on their future work relating to this point.

Reviewer 1’s citations and additional suggestions have been included on page 11, line 192 in the Task Design section of the manuscript and page 37, line 523 in the Discussion section of the manuscript. In the task design, it is now explained that the monetary distributions were selected to conform to universal scaling parameters of the PDG such that the Dg’ and the Dr’ are equal and greater 1, generating a Donor & Recipient style template that incentives defection at no cost to the defector given a single decision but conversely incentivizes cooperation in an iterated game but at a cost to the cooperator. Additionally, in the discussion section, we speculate about the results of a deviation away from the standard dilemma strength of Dg’ = Dr’ = 1 on the behavioral and neural responses of our participants, suggesting as a likely outcome that strengthening the paradigm would heavily incentivize defection and elicit the kind of reward-based neural activity we had originally predicted over the course of the game.

Reviewer #2

1) Figure captions are too short, in order make it more readable, please add one or two sentences to conclude its main conclusions. Please note that not all the readers have patients to read the whole text, add this would help the reader immensely.

The captions for each figure in the manuscript have been expanded and elaborated upon. The additional content will hopefully clarify the nature of the illustrations for both the reviewers and well as prospective readers, particularly for the contrast map illustrations which we do acknowledge were not appropriately addressed beforehand.

2) Page 11, line183, and page 13, line 230-236 rhs. There are 40 rounds in human games and 20 rounds in computer games, I am not clear why the authors organized three treatments rather than four? Please note that when you calculate the cooperation rate in human and computer games, the baseline is different. Please clarify and expand.

Reviewer 2 was rightly concerned with the lack of balance in the treatment conditions between human and computer games in the present study. The experimental design was originally organized with a primary intention of 1) isolating BOLD activity associated with human gameplay and 2) avoiding the fatigue effects that oftentimes result when the participant has spent an extended period of time in the scanner. The entire session protocol spanned almost an hour and in the experience of 2 co-authors who tried to add additional games previously, the degree of effort and attention placed on the task dropped substantially after 3 games. With this caveat in mind, the computer condition was simply included in the design as an exploratory, experimental variable and not as an effective control condition to serve as a basis of comparison, which was not required for our task-based fMRI analysis.

3) Although the present paper is greatly different with West’s paper, I think there are some connections. In West’s paper, participants are unaware of its opponent’s information although they play with human, they just know that they input one number and will get a reward. In your paper, when participants play in ‘human’ games, the information of its opponent’s was almost complete, but the participants were wrongly thinking its opponents are human. If possible, please consider the connections and expand.

The paper the reviewer is referencing utilizes the public goods game as opposed to the Prisoner’ Dilemma as a model of reciprocity. In the public goods game, a group of four individually decides how much money (up to 40 virtual coins) they will donate to the “pot” (public good), which is then evenly distributed between all players. The experimental condition the reviewer refers to is called the black box condition. In this condition, the players only know their decision and what they earn, but they do not know that they are playing with other people and what their co-players donates or their earnings, while in our PD game the decision and earnings of the opponent are known, but the participant is deceived to believe they are playing a human-being when they are actually playing a computerized algorithm.

The levels of “cooperation”, or donation in the black box condition of the public goods game was similar to the standard version of the game in which all information is known except for the earnings of others within the group. However, when the earnings information of the partners was known (enhanced condition), donations were reduced rather than increased, contradicting prosocial assumptions made about humans in these kind of mixed-motive dilemmas. 

In our current study, behavior corresponded with what has been seen in previous literature; gameplay with ostensible humans was associated with increased levels of cooperation in comparison to gameplay with computers. These results contradict the findings of the West paper by supporting the prosocial hypothesis that is oftentimes assumed to be underlying these game-theoretic paradigms.

I believe that the nature of the paradigms themselves and constraints of the conditions explain these differences. In the black box condition of the public goods, participants did not even know they were playing with other people. In this case, it would be counterintuitive to believe that a contribution of nothing would enrich themselves if they are only playing with themselves, therefore the only logical decision is to donate some tokens and cooperate. Once more knowledge is revealed concerning the constraints of the public goods game (standard and enhanced condition) it is easy to see that “defecting” or refusing to donate is a favorable option since you have 3 other partners you can depend upon to donate to the pot. In our Prisoner’s Dilemma game, increased knowledge of the parameters of the game does not in fact lead to increases in defection because in an iterated format, prolonged defection diminishing overall earning potential for both parties and the participant only has one partner to rely upon to secure their purse. Therefore, cooperation is the favorable strategy in comparison to the Public Goods game. Additionally, let’s say we ran our PD game with a black box constraint so that the participant only knew they could cooperate or defect, and that they would earn some money. While the strategy here seemingly becomes much more complicated because of the variety of outcomes in comparison to the public goods game, ultimately, I believe participants would favor defection because you can at least avoid a non-zero outcome this way (unreciprocated cooperation). In conclusion, the parameters of the PD and public goods game support deviating strategies due to the nature and assumptions of the paradigms.

---

## [Decision Letter · Decision Letter 1]

18 Feb 2021

The Prisoner’s Dilemma paradigm provides a neurobiological framework for the social decision cascade

PONE-D-20-41107R1

Dear Dr. Thompson,

We’re pleased to inform you that your manuscript has been judged scientifically suitable for publication and will be formally accepted for publication once it meets all outstanding technical requirements.

Kind regards,

Jun Tanimoto

Academic Editor

PLOS ONE

Additional Editor Comments (optional):

Reviewers' comments:

Reviewer's Responses to Questions

**Comments to the Author**

1. If the authors have adequately addressed your comments raised in a previous round of review and you feel that this manuscript is now acceptable for publication, you may indicate that here to bypass the “Comments to the Author” section, enter your conflict of interest statement in the “Confidential to Editor” section, and submit your "Accept" recommendation.

Reviewer #1: (No Response)

Reviewer #2: (No Response)

2. Is the manuscript technically sound, and do the data support the conclusions?

Reviewer #1: Yes

Reviewer #2: Yes

3. Has the statistical analysis been performed appropriately and rigorously? 

Reviewer #1: Yes

Reviewer #2: Yes

4. Have the authors made all data underlying the findings in their manuscript fully available?

Reviewer #1: Yes

Reviewer #2: Yes

5. Is the manuscript presented in an intelligible fashion and written in standard English?

Reviewer #1: Yes

Reviewer #2: Yes

6. Review Comments to the Author

Reviewer #1: The revised MS seems adequate for publication. The authors deliberately solved all questions I suggested.

Reviewer #2: (No Response)

7. PLOS authors have the option to publish the peer review history of their article (what does this mean?). If published, this will include your full peer review and any attached files.

Reviewer #1: No

Reviewer #2: No

---

## [Editor Report · Acceptance letter]

23 Feb 2021

PONE-D-20-41107R1 

The Prisoner’s Dilemma paradigm provides a neurobiological framework for the social decision cascade 

Dear Dr. Thompson:

I'm pleased to inform you that your manuscript has been deemed suitable for publication in PLOS ONE. Congratulations! Your manuscript is now with our production department. 

Kind regards, 

on behalf of

Prof. Jun Tanimoto 

Academic Editor

PLOS ONE